# Simultaneous Determination of Nine Quinolones in Pure Milk Using PFSPE-HPLC-MS/MS with PS-PAN Nanofibers as a Sorbent

**DOI:** 10.3390/foods11131843

**Published:** 2022-06-22

**Authors:** Lanlan Wei, Yanan Chen, Dongliang Shao, Jingjun Li

**Affiliations:** 1College of Food Engineering, Anhui Science and Technology University, Chuzhou 233100, China; 18356130656@163.com (L.W.); ynchen16@163.com (Y.C.); 2Anhui Guoke Testing Technology Co., Ltd., Hefei 230000, China; 17855348740@163.com

**Keywords:** polystyrene-polyacrylonitrile nanofibers, quinolones, packed-fiber solid-phase extraction, HPLC-MS/MS, pure liquid milk

## Abstract

In this study, a packed-fiber solid-phase extraction (PFSPE)-based method was developed to simultaneously detect nine quinolones, including enrofloxacin (ENR), ciprofloxacin (CIP), ofloxacin (OFL), pefloxacin (PEF), lomefloxacin (LOM), norfloxacin (NOR), sarafloxacin (SAR), danofloxacin (DAN), and difloxacin (DIF), in pure milk, using high-performance liquid chromatography coupled with tandem mass spectrometry (HPLC-MS/MS). Polystyrene (PS) and polyacrylonitrile (PAN) were combined to form PS-PAN composite nanofibers through electrospinning. The nanofibers were used to prepare the home-made extraction columns, and the process was optimized and validated using blank pure milk. The analytical method showed high accuracy, and the recoveries were 88.68–97.63%. Intra-day and inter-day relative standard deviations were in the ranges of 1.11–6.77% and 2.26–7.17%, respectively. In addition, the developed method showed good linearity (R^2^ ≥ 0.995) and low method quantification limits for the nine quinolones (between 1.0–100 ng/mL) for all samples studied. The nine quinolones in the complex matrix were directly extracted using 4.0 mg of PS-PAN composite nanofibers as a sorbent and completely eluted in 100 μL elution solvent. Therefore, the developed PFSPE-HPLC-MS/MS is a sensitive and cost-effective technique that can effectively detect and control nine quinolones in dairy products.

## 1. Introduction

Milk is a nutritious liquid food produced by animals that is a key nutritional source for humans. Milk production also significantly contributes to the global economy. Antibiotic treatments protect animals from infections such as mastitis [1]. However, inappropriate use of antibiotics such as quinolones can deteriorate the quality of milk and other dairy products. Since quinolones are effectively used to resist Gram-positive and Gram-negative bacteria, they are widely applied for therapeutic purposes [2]. Animal products may contain some fractions of antibiotics due to excessive antibiotic treatment. The antibiotic remaining in the animal products can potentially have severe consequences on human health via the food chain [3]. Thus, monitoring the residues of quinolones in dairy products is vital to human health. The control of veterinary drug residues is an important measure for ensuring consumer protection [4]. Therefore, an efficient and sensitive method for detecting quinolones should be developed and validated to monitor the residues in food, ensuring the healthy consumption of dairy products.

A variety of analytical methods such as diode array UV [5,6], fluorescence [7,8], mass spectrometry (MS) [9,10,11], and capillary electrophoresis (CE) [5,12] have been applied to determine the quinolones in dairy products. Due to the complex matrices and low fluoroquinolone (FQ) concentrations in food samples, suitable sample pretreatment is usually required to remove interfering matrix components and enrich the analytes to appropriate concentrations for subsequent analysis, irrespective of the analytical method selected. To reduce matrix effects (ME), remove interferences, and improve the efficiency of the method [13], the samples are pretreated using various extraction techniques, including solid-phase extraction (SPE) [5,6,7], liquid–liquid extraction (LLE) [2], and online SPE [10], to extract target analytes with different solvents. However, conventional extraction methods such as LLE [14,15] and SPE [16] suffer from low selectivity or a long analysis time, because solvent evaporation to dryness is required. Moreover, large volumes of organic solvents are consumed, which are harmful to human health.

Packed-fiber solid-phase extraction (PFSPE), as a novel SPE technique, could simplify the process of sample preparation. Purification, concentration, and desorption could be undertaken in one step, without using a membrane filter, heating for evaporation, and drying in a nitrogen stream. In addition to the above features, electrospun nanofibers use a smaller sorbent-bed mass due to their unique physicochemical properties (such as a large surface-area-to-volume ratio), reducing the quantity of organic solvents and increasing extraction efficiencies for trace analyses [17]. Thus, electrospun nanofibers are a good potential sorbent material for SPE-based techniques [18,19]. In many research studies, polystyrene/polypyrrole (PS/PPy) nanofibers have become one of the most promising materials for potential applications [20]. PPy/PAN nanofibers have been used as extraction materials for cationic dyes in wastewater [21].

In this study, polyacrylonitrile-polystyrene (PAN-PS) composite nanofibers (a selective sorbent) were prepared and characterized. A PFSPE-based method, with PAN-PS nanofibers (as the adsorbent) and high-performance liquid chromatography tandem mass spectrometry (HPLC-MS/MS), was used to detect nine quinolones, namely, enrofloxacin (ENR), ciprofloxacin (CIP), ofloxacin (OFL), pefloxacin (PEF), lomefloxacin (LOM), norfloxacin (NOR), sarafloxacin (SAR), danofloxacin (DAN), and difloxacin (DIF), in pure milk samples. In addition, this method was used to compare the extraction efficiencies of PAN and PAN-PS nanofibers. Other important parameters influencing the extraction efficiency, such as the concentration of eluant, the amount of adsorbent, the ionic strength, and the reusability of the PAN-PS nanofibers, were investigated and optimized. Finally, the method was used to detect nine quinolones in plain milk.

## 2. Materials and Methods

### 2.1. Materials and Reagents

Analytical grade chemicals and reagents were used. N,N-Dimethylformamide (DMF), poly-acrylonitrile (PAN, retained molecular weight: 150,000 Da), polystyrene (PS, retained molecular weight: 192,000 Da), enrofloxacin (ENR), ciprofloxacin (CIP), ofloxacin (OFL), pefloxacin (PEF), lomefloxacin (LOM), norfloxacin (NOR), sarafloxacin (SAR), danofloxacin (DAN), and difloxacin (DIF) were purchased from Shanghai Macklin Biochemical company (Shanghai, China, www.macklin.cn.qianyan.biz (accessed on: 11 May 2022)). Methanol (CH_3_OH, HPLC-grade), acetonitrile (CH_3_CN, HPLC-grade), formic acid (HCOOH, HPLC-grade), sulfuric acid (H_2_SO_4_), sodium chloride (NaCl), and EDTA–McIlvaine buffer (pH4.0) were all purchased from Aladdin company (Shenzhen, China, www.aladdin-e.com (accessed on: 11 May 2022)).

Single stock solutions of 0.1 mg/mL of the nine quinolones, namely, ENR, CIP, OFL, PEF, LOM, NOR, SAR, DAN, and DIF in methanol were separately prepared. The working mixture solutions were prepared via appropriate dilution of the stock solutions with 1.0 ng/mL, 5.0 ng/mL, 20.0 ng/mL, 50.0 ng/mL, and 100.0 ng/mL of pure water at 4 °C in a volumetric flask.

### 2.2. Equipment and HPLC-MS-MS Conditions

The separation of the nine quinolones from the pure milk extracts was performed using an HPLC system (Agilent 1200 Series HPLC system, Agilent Technologies, Santa Clara, CA, USA) placed in an Anhui Guoke Testing Technology Co., Ltd. (www.guoketest.com.cn (accessed on: 11 May 2022)) system consisting of a vacuum degasser, an autosampler, and a binary pump, equipped with a reversed-phase SB-C18 analytical column (2.1 × 100 mm, 2.7 μm particle size, Agilent Technologies, Santa Clara, CA, USA). The mobile phase consisted of 0.1% of formic acid in water (A) and acetonitrile (B). Gradient conditions such as 0–1.0 min (95% A), 1.1–4.0 min (2% A), and 4.1–6.0 min (95% A) were set up, and the flow rate was maintained at 0.3 mL/min.

The mass spectrometric acquisition was performed on an Agilent 6460 triple quadrupole mass spectrometer (Agilent Technologies, Palo Alto, CA, USA) equipped with an electrospray ionization source to generate positive ions [M + H]^+^. In order to achieve the best ionization of the nine quinolones, an ion spray voltage of 4.0 kV in the positive mode was employed. The nebulizing gas was nitrogen (99.999%), and the rate was 7.0 L/min. The temperature of the symmetrical heaters was 300 °C, and the flow rate of the sheath gas was 11 L/min. Finally, the injection volume was 2 μL. The retention time, precursor ion, ionic product, and collision energy for each analyte are listed in Table 1.

### 2.3. Preparation of PS-PAN Nanofibers

PS-PAN nanofibers were fabricated as follows. First, PAN (15%, *w*/*v*) solution was prepared by dissolving an appropriate amount of PAN in the DMF solvent to form a polymer solution. PS (5%, *w*/*v*) was added to the polymer solution after the PAN was completely dissolved. The mixed electrospinning solution was stirred at 6000 rpm at an ambient temperature of 23~25 °C. Then, it was loaded into the glass syringe, which was equipped with a steel needle with a tip diameter of 0.33 mm. Finally, the conditions for electrospinning were as follows: an anodic voltage of 22 kV, a distance between the tip of the needle and the collector of 15 cm, and a mixed electrospinning solution flow rate of 2.0 mL/h. The nanofibers were collected on the collector and dried at room temperature.

### 2.4. Characterization

The surface morphologies of the electrospun fibers were observed using a scanning electron microscope (SEM, Supra55, ZEISS, Jena, Germany) at 20 kV. The average diameter of the electrospun fibers was determined by analyzing 30 single fibers using the Nano Measurer software, version 1.2 (Nanjing, China). The molecular structure of the nanofiber membrane was confirmed using ATR-IR spectroscopy (FTIR, NicoletiS10, Thermo Fisher, Waltham, MA, USA), carried out in the 4000–500 cm^−1^ range. The X-ray diffraction analysis was carried out using an X-ray diffractometer (XRD, XD-3X, Persee General, Beijing, China), and the XRD spectra were recorded over a 2θ range of 5° to 60°, with a step of 0.02°.

### 2.5. Sample Pretreatment

Pure milk (150 mL, Yili) was purchased from a local supermarket in Chuzhou, China. According to the method described in the literature, one sample was used as the blank sample without quinolones [22]. The preparation of the milk sample included the following steps: (i) 2.0 mL milk was added into a 50 mL centrifuge tube; (ii) nine quinolones (ENR, CIP, OFL, PEF, LOM, NOR, SAR, DAN, and DIF) with appropriate concentrations (2 ng/mL, 10 ng/mL, and 25 ng/mL) were added to the sample; (iii) 10.0 mL of EDTA–McIlvaine buffer solution (pH 4.0) was added to the mixed solution; (iv) the mixture was sonicated for 20 min and centrifugated at 10,000 rpm for 30 min before the supernatant was separated; and (v) the total supernatant was transferred to a clean tube and stored at 4 °C before the pretreatment using PFSPE.

To effectively implement the extraction, 4.0 mg of nanofibers was packed into the PFSPE column at the tip end of the storage cartridge [23]. Then, the column was activated using 200 μL of methanol and 200 μL of deionized water. A 500 mL volume of supernatant was loaded into the gas-tight plastic syringe and then pushed through the sorbent under atmospheric pressure. Targets adsorbed on the nanofibers were eluted using 100 μL of eluent, as shown in Figure 1. Hence, the extraction recovery was calculated following Equation (1):q = (A_1_/5/A_0_) × 100%(1)
where A_0_ is the peak area of 10 μL of the nine quinolones standard solution and A_1_ is the peak area of 10 μL of the nine quinolones eluent solution.

### 2.6. Method Validation

Different standard solutions were prepared and verified through PFSPE-HPLC-MS/MS to validate the linearity, precision, detection limit (LOD), and quantification (LOQ) limit under optimal conditions. The linearity was determined by analyzing standard solutions ranging from 1.0 to 100 ng/mL in blank pure milk. The standard solution with the lowest concentration (2.0 ng/mL) of the nine quinolones was analyzed to obtain the LOD and LOQ, which were calculated for signal-to-noise ratios (S/N) of 3 and 10, respectively. The relative standard deviation (% RSD) was used as the precision and calculated according to the reproducibility (intra-day precision, *n* = 3 and inter-day precision, *n* = 6). In this study, the matrix-matched slope divided by the solvent-based slope equals the ME, and the formula is:((slope matrix − matched/slope solvent) *×* 100).

## 3. Results and Discussion

### 3.1. Characterization of Nanofibers

Compared with PAN nanofibers, no significant morphological change was observed in the SEM images of PAN-PS nanofibers (Figure 2a,b). In addition, the rough surface of the PAN-PS nanofibers, compared with PAN nanofibers, remained unchanged. The diameter size distributions of different PAN-PS and PAN nanofibers are shown in Figure 2a’,b’. The mean diameters of the PAN and PAN-PS nanofibers were 0.6 μm and 0.9 μm, respectively. The nanofibers were uniform and dense with a network structure. However, there were few beads and several beads on PAN nanofibers and PAN-PS nanofibers, respectively.

### 3.2. Molecular Interactions of Nanofibers

The Fourier transform infrared spectrometry (FTIR) spectra of the PAN and PAN/PS nanofibers are shown in Figure 3. The characteristic peaks of PAN nanofibers were composed of conjugated C−N stretching at 2242 cm^−1^ and −CH_2_− stretching vibrations at 1450 cm^−^^1^ and 2491 cm^−1^. After blending the PS, as shown in the spectrum for PAN/PS nanofibers, many new peaks appeared in the IR spectra of the PAN/PS nanofibers. The new peaks at 698 cm^−1^ and 756 cm^−1^ were absorption peaks due to C−H vibrations in the PS benzene ring. The above results indicate that the combination of PAN and PS does not produce new chemical bonds but exists as PAN/PS composite nanofibers.

### 3.3. X-ray Diffraction Analysis (XRD)

X-ray diffractograms of the PAN nanofibers, PS nanofibers, and PAN/PS nanofibers are shown in Figure 4. The broad diffraction peaks at 2θ = 10° and 20° are the diffraction peaks of PS nanofibers. The sharp diffraction peaks at 2θ = 16° and 26° indicate the highly crystalline nature of PAN nanofibers. The XRD pattern of PAN/PS nanofiber membranes exhibited a similar pattern to the PAN nanofibers and PS nanofibers. Sharp diffraction peaks were observed at 2θ = 16° and 26°. The broad peak at 2θ = 10°, corresponding to PS, did not appear in the XRD pattern of PAN/PS nanofibers. Moreover, the lower intensity of the peaks at 2θ = 16° and 26° suggested that the overall crystallinity of the PAN/PS nanofibers was slightly lower than that of PS but higher than that of PAN nanofibers. The results showed that pure PAN and PS particles were physically mixed and generated strong molecular interactions.

### 3.4. Comparison of Adsorption Efficiencies between PAN Nanofibers and PAN/PS Composite Nanofibers

First, the effect of the nanofiber characteristics on the efficiency of extraction was investigated. In this study, the extraction recoveries of the two types of nanofibers prepared with different frameworks were evaluated. The extraction recoveries of PAN/PS composite nanofibers for nine quinolones (ENR, CIP, OFL, PEF, LOM, NOR, SAR, DAN, and DIF) were 97.5%, 99.2%, 93.2%, 98.8%, 93.2%, 98.3%, 87.6%, 98.6%, and 89.1%, respectively, and these values were much higher than those for pure PAN nanofibers. The target analytes were difficult to detect due to the polymeric nature of PAN and the existence of strong chemical bonds between its molecules. Moreover, the result could be explained using the interaction mechanism between PAN-PS and the analytes, due to the conjugated π–π structure in the PS backbone, which showed a strong interaction between the analytes. A hydrogen bonding interaction could also exist between the polymer and analytes.

After extraction, quinolones were desorbed from the sorbents using eluent prior to HPLC-MS/MS analysis. To determine the optimal elution solvent, 100 μL of 70% (*v*/*v*) methanol, 80% (*v*/*v*) methanol, 90% (*v*/*v*) methanol, and 1% (*v*/*v*) sulfuric acid/70% methanol were evaluated (Figure 5b). Low recoveries were observed when the various concentrations of methanol were used, but an increase in recovery was observed after 1% sulfuric acid was added (Figure 5c). Compared with the desorption ability of 70% methanol, 1% formic acid/70% methanol exhibited higher recovery rates (82.4–100.7%).

The amount of PAN-PS nanofibers played a vital role in the recovery of the nine quinolones. If there are insufficient nanofibers, the target analytes cannot be completely adsorbed; however, excessive nanofibers may lead to incomplete elution. As shown in Figure 5d, as the amount of nanofibers increased from 3.0 mg to 4.0 mg, the recovery of the nine targets increased significantly. However, for amounts of nanofibers ranging from 5.0 mg to 8.0 mg, the elution rates of the nine targets decreased significantly. Therefore, 4.0 mg of 15% polyacrylonitrile and 5% polystyrene composite nanofibers was selected for solid-phase extraction in this study.

To explore the effect of salt addition on the extraction of the nine quinolones, different concentrations (0%, 2%, 4%, 6%, 8%, and 10.0% *w*/*v* of the analytes) were added to the samples. The addition of salt to the samples reduced the solubility of the analytes and the amount of the analytes extracted [24]. Figure 5e shows that there were no significant changes in extraction efficiency. Hence, adjustment of the ionic strength was not necessary in this study.

To exhibit the reproducibility of the synthesis of sorbent, the extraction recoveries of the nine quinolones were tested. As shown in Figure 5f, the extraction recoveries were almost the same under the same conditions of synthesis and pretreatment of the nine quinolones. The results indicate that the PAN/PS composite nanofibers exhibited good chemical and mechanical stability and reproducibility.

### 3.5. Method Validation

The reproducibility, linearity, and LOD were investigated under optimal conditions. Milk samples were randomly obtained from a local market in Chuzhou (China). According to the national standard method, one of the milk samples without the nine quinolones was selected as a blank for calibration and validation.

Under optimum conditions, everyday snack samples spiked with the nine quinolones were employed for analyzing the methodological parameters. As shown in Table 2, good linearity was found in the range of 1.0−100 ng/mL for the nine quinolones. The coefficients of correlation (R^2^) ranged from 0.9992 to 0.9999. The LODs of all analytes were in the range of 0.16–0.39 ng/mL, and the LOQs were in the range of 0.53−1.29 ng/mL, indicating that the method can effectively determine the nine quinolones in plain milk samples. As shown in Table 2 and Table 3, a high efficiency of extraction was found for all analytes in the pure milk sample. Recoveries were obtained over a reasonable range from 88.68% to 97.63% for the nine analytes in the spiked concentration samples. The intra-day range for the three spiked concentration samples was from 1.11% to 6.77%, and the inter-day RSD was in the range 2.26−7.17%. All these values are below 12%.

### 3.6. Matrix Effect

The extraction and analysis of the nine quinolones in pure milk could be affected by other factors. If the value of the matrix effect (ME) is 100%, this shows that there is no interference from the matrix; however, if the value is more than 100%, it shows that the ME enhances the analysis signal. Finally, if the value is less than 100%, the ME reduces the signal [25]. The results for the ME were evaluated and are summarized in Table 4. Although the ME values of the nine analytes were less than 100%, they all ranged from 94.2% to 98.3%. Therefore, the effect of matrix interference on the analysis signal of pure milk was insignificant.

### 3.7. Comparison of This Method with Other Measurement Methods

The advantages of the developed method for detecting the nine quinolones in plain milk are summarized in Table 5. The developed method was compared with various preparation methods in the existing literature. Firstly, the fabricated PS-PAN nanofibers were used to analyze the nine quinolones in plain milk for the first time. Secondly, the sample pretreatment required no evaporative drying with nitrogen or treatment of the filter membrane. Generally, the procedure is simplified and easy to master.

### 3.8. Application to Real Samples

The samples were then spiked with ENR, CIP, OFL, PEF, LOM, NOR, SAR, DAN, and DIF standards at 50 ng/mL levels, to assess the matrix effects. Nine quinolones were detected in milk samples, as shown in Figure 6. This results show that the developed method can effectively detect quinolones in samples where they cannot be detected using the standard method.

## 4. Conclusions

The PFSPE-based PAN-PS composite nanofibers were used as sorbents to simultaneously extract and analyze nine quinolones (ENR, CIP, OFL, PEF, LOM, NOR, SAR, DAN, and DIF) in pure milk, using HPLC-MS/MS. After the optimization of the process, the method showed good linearity, precision, LOD, and LOQ, with high accuracy, and low ME values. The analytical method showed high accuracy, with recoveries of 88.68−97.63%. Intra-day and inter-day relative standard deviations were in the ranges of 1.11−6.77% and 2.26−7.17%, respectively. In addition, the developed method showed good linearity (R^2^ ≥ 0.995) and low method quantification limits for the nine quinolones (between 1.0−100 ng/mL) in all samples studied. Moreover, compared with the sample pretreatment in conventional methods, the sample pretreatment in the developed method required no evaporative drying under nitrogen or filter membrane treatment. The results show that PFSPE-HPLC−MS/MS is a sensitive and cost−effective technique for detecting nine quinolones in plain milk. However, there are also some limitations. For example, the column packing is not automated. If it could be automated, the consistency of detection would be improved considerably.

## Figures and Tables

**Figure 1 foods-11-01843-f001:**
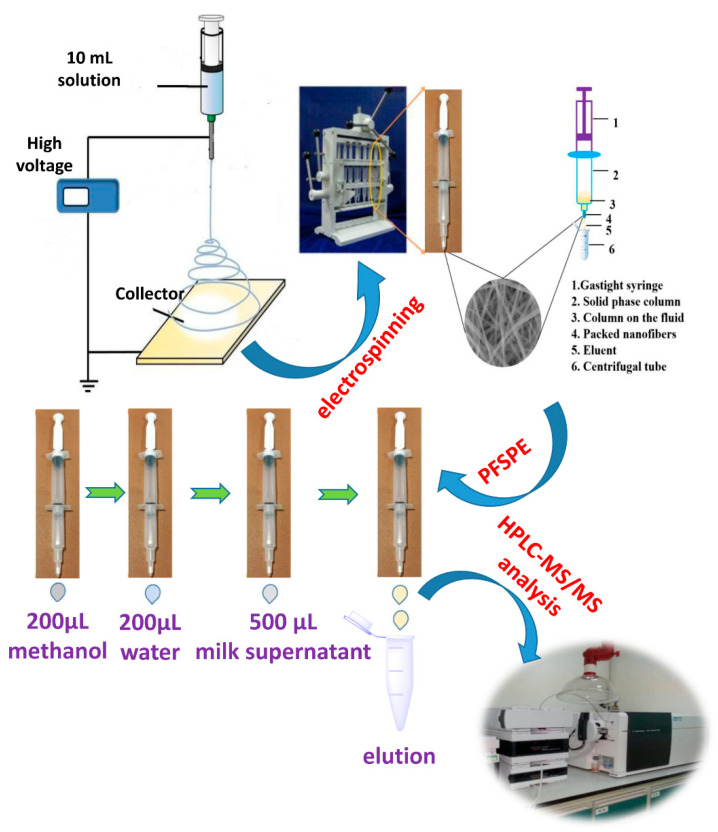
Schematic of packed-nanofiber solid-phase extraction process.

**Figure 2 foods-11-01843-f002:**
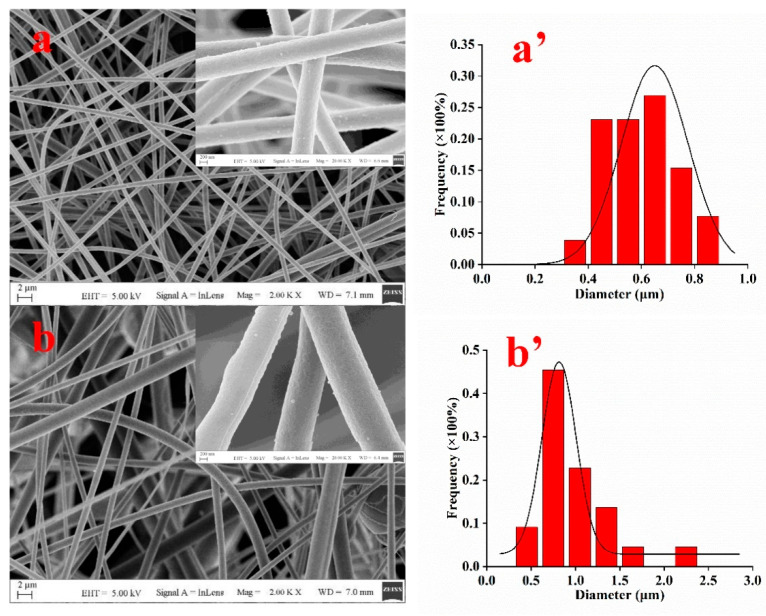
The SEM images of (**a**) PAN nanofibers and (**b**) PAN-PS nanofibers and the size distributions of (**a’**) PAN nanofibers and (**b’**) PAN-PS nanofibers.

**Figure 3 foods-11-01843-f003:**
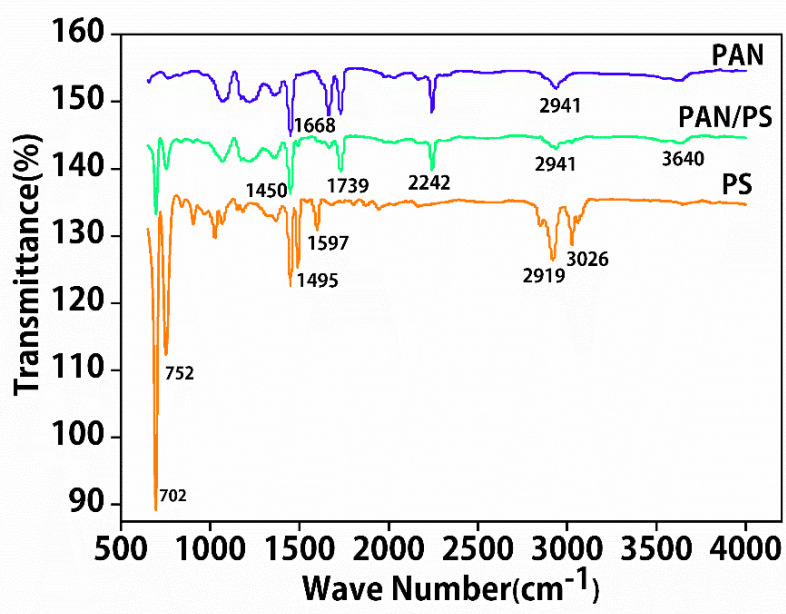
FTIR spectra of PAN nanofibers and PAN/PS composite nanofibers.

**Figure 4 foods-11-01843-f004:**
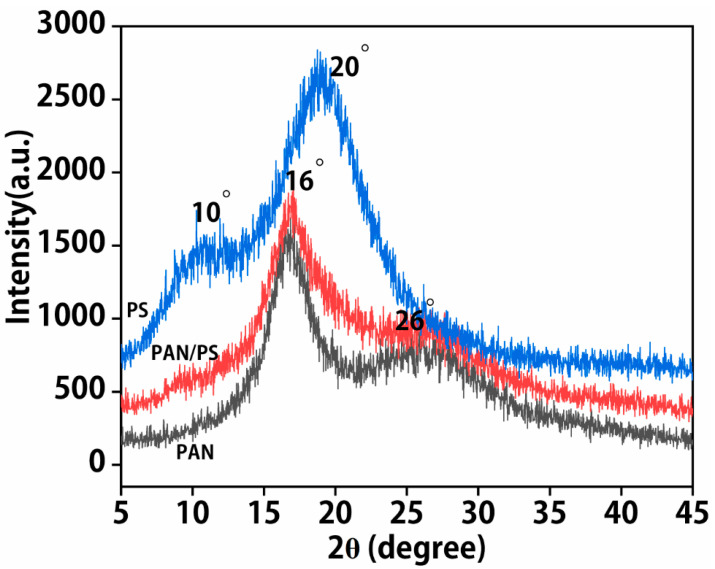
XRD patterns of PAN nanofibers, PS nanofibers, and PAN/PS composite nanofibers.

**Figure 5 foods-11-01843-f005:**
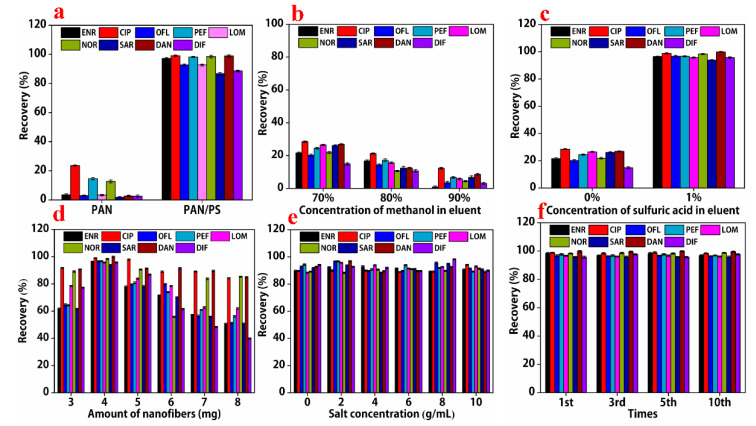
(**a**) Adsorption efficiency of nine quinolones onto PAN and PAN-PS nanofiber extraction columns; (**b**–**e**) effects of methanol concentration in the eluent, sulfuric acid concentration in the eluent, the amount of nanofibers, and salt concentration on the extraction efficiency at a concentration of 50 ng/mL for nine quinolones; (**f**) effect of the run number for reproducibility on extraction efficiency at the concentration of 50 ng/mL for nine quinolones.

**Figure 6 foods-11-01843-f006:**
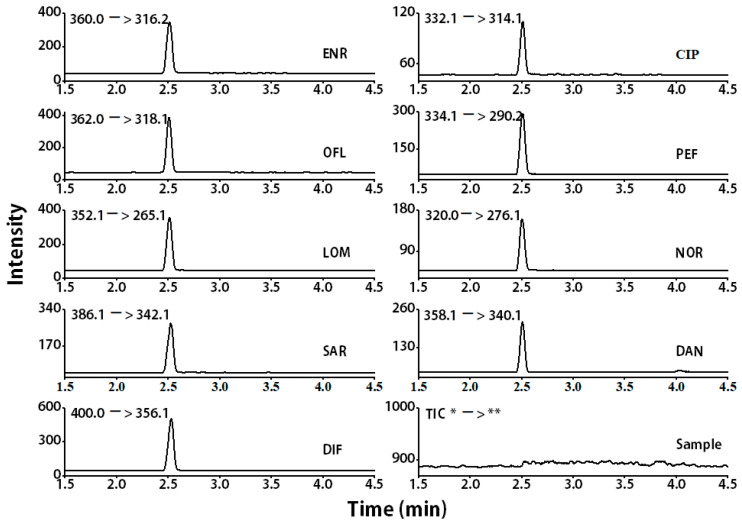
MS/MS spectra of all analytes in a spiked pure milk sample (spike concentration: 5.0 ng/mL). * Precursor ion, ** Product ions.

**Table 1 foods-11-01843-t001:** Compound-dependent MS parameters for each analyte.

Analyte	Structure	Formula	Precursor ion (*m*/*z*)	Product ions (*m*/*z*)	Cone Voltage (V)	Collision Energy (V)
Enrofloxacin (ENR)	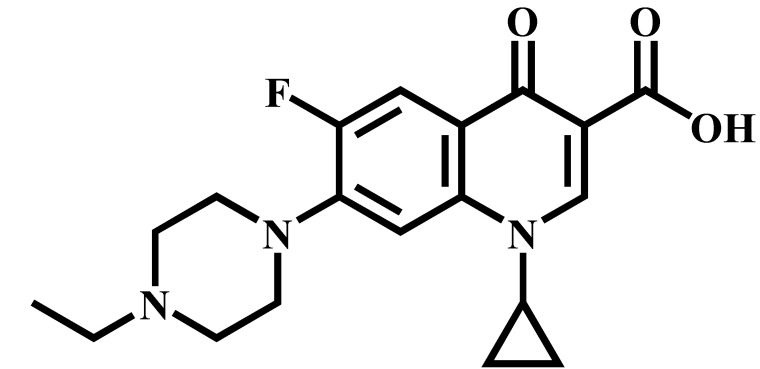	C_19_H_22_FN_3_O_3_	360.0	316.2 *	120	20
				342.1		20
				244.9		40
Ciprofloxacin (CIP)	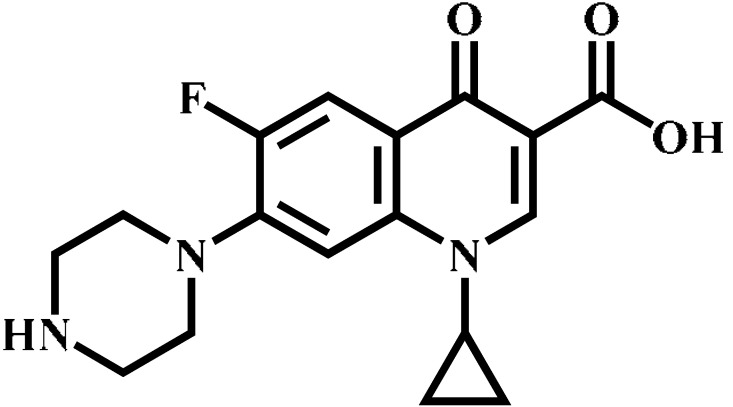	C_17_H_18_FN_3_O_3_	332.1	314.1 *	135	20
				231.0		42
Ofloxacin (OFL)	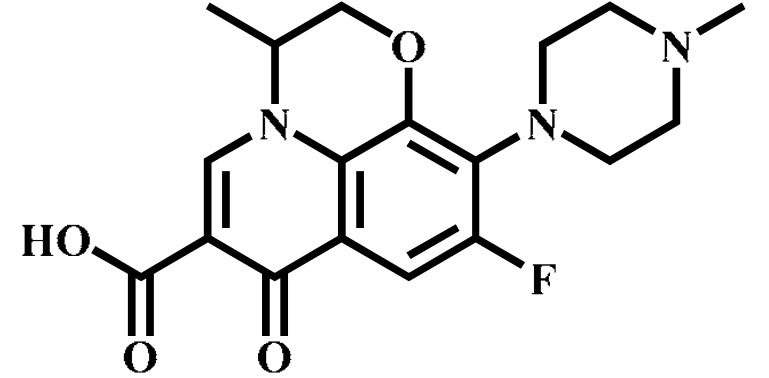	C_18_H_20_FN_3_O_4_	362.0	318.1 *	130	15
				261.1		26
Pefloxacin (PEF)	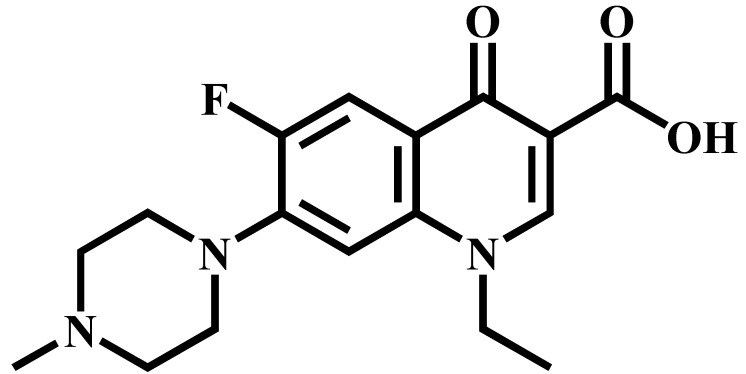	C_17_H_20_FN_3_O_3_	334.1	290.2 *	130	16
				316.2		20
Lomefloxacin (LOM)	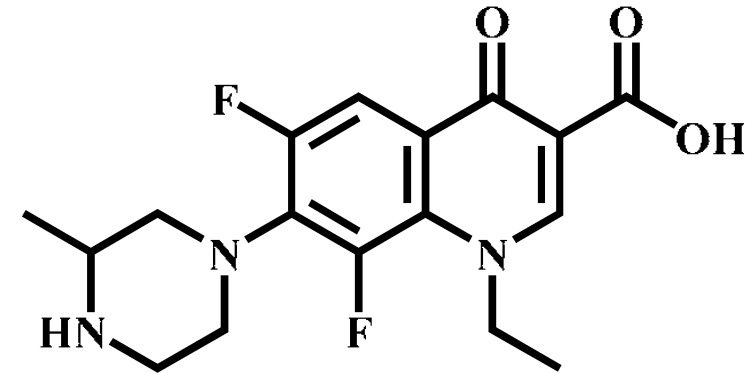	C_17_H_19_F_2_N_3_O_3_	352.1	265.1 *	130	20
				308.1		10
Norfloxacin (NOR)	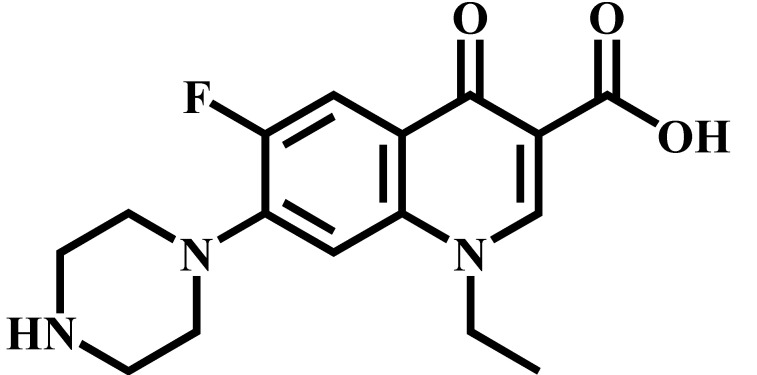	C_16_H_18_FN_3_O_3_	320.0	276.1 *	130	15
				302.1		20
Sarafloxacin (SAR)	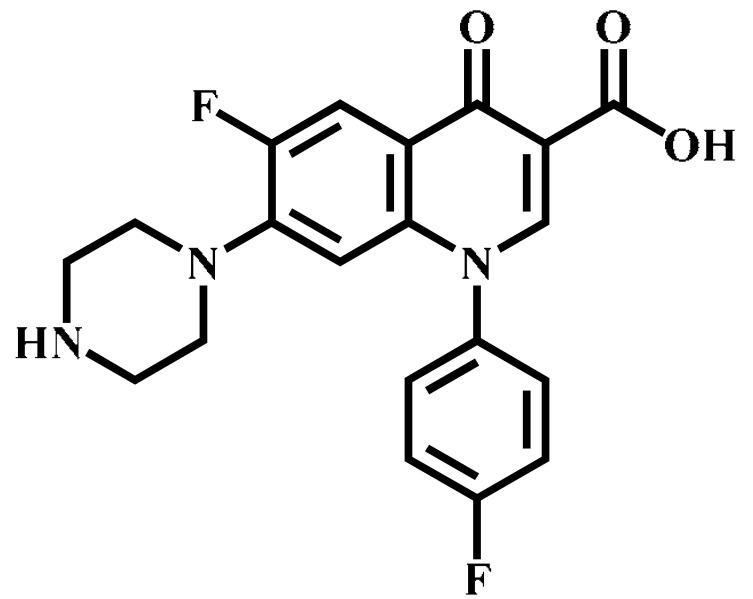	C_20_H_17_F_2_N_3_O_3_	386.1	342.1 *	130	15
				368.1		20
Danfloxacin (DAN)	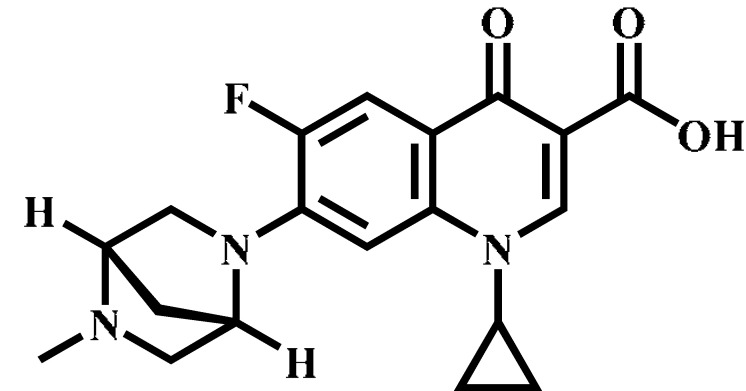	C_19_H_20_FN_3_O_3_	358.1	340.1 *	140	25
				255.0		46
Difloxacin (DIF)	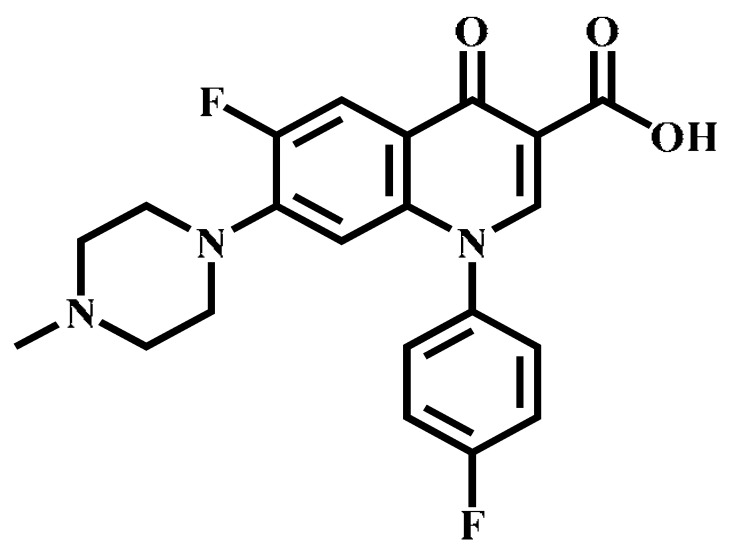	C_21_H_19_F_2_N_3_O_3_	400.0	356.1 *	140	20
				382.1		20

* Ions used for quantification.

**Table 2 foods-11-01843-t002:** Performance of the method.

Analyte	Linearity Range (ng/mL)	Correlation Coefficient (R^2^)	LOD (ng/mL)	LOQ (ng/mL)
ENR	1.0–100	0.9999	0.18	0.59
CIP		0.9994	0.36	1.20
OFL		0.9996	0.23	0.76
PEF		0.9996	0.16	0.53
LOM		0.9997	0.23	0.75
NOR		0.9994	0.39	1.29
SAR		0.9996	0.27	0.91
DAN		0.9992	0.18	0.59
DIF		0.9998	0.36	1.19

**Table 3 foods-11-01843-t003:** Absolute recoveries of nine quinolones from pure milk samples via PFSPE−HPLC−MS/MS analysis.

Analyte	Spiked Concentration (ng/mL)	Recovery (%) ± RSD (*n* = 3)	
		Intra−Day	Inter−Day
ENR	2	92.51 ± 1.71	90.64 ± 5.02
	10	90.94 ± 2.61	88.68 ± 3.10
	25	95.39 ± 4.41	93.25 ± 4.67
CIP	2	97.02 ± 1.56	96.72 ± 3.92
	10	97.27 ± 5.38	95.59 ± 6.07
	25	93.56 ± 2.67	91.30 ± 4.16
OFL	2	97.93 ± 2.74	97.64 ± 3.15
	10	95.53 ± 3.15	94.20 ± 2.61
	25	94.92 ± 2.27	95.94 ± 2.01
PEF	2	93.51 ± 1.60	93.12 ± 3.34
	10	92.60 ± 1.08	90.74 ± 1.10
	25	89.86 ± 1.53	89.10 ± 3.50
LOM	2	94.11 ± 2.36	91.12 ± 5.01
	10	95.78 ± 1.68	93.30 ± 4.22
	25	92.22 ± 2.46	92.84 ± 2.96
NOR	2	91.81 ± 1.20	91.49 ± 2.30
	10	92.28 ± 2.66	92.12 ± 3.94
	25	95.26 ± 1.28	92.86 ± 2.41
SAR	2	90.98 ± 2.36	90.86 ± 2.06
	10	92.06 ± 3.77	91.72 ± 2.37
	25	91.49 ± 2.55	92.04 ± 2.93
DAN	2	96.00 ± 2.66	94.39 ± 4.29
	10	91.32 ± 1.25	90.49 ± 3.06
	25	97.63 ± 2.40	94.96 ± 3.09
DIF	2	95.33 ± 2.04	94.11 ± 4.35
	10	94.61 ± 3.05	94.31 ± 2.72
	25	95.98 ± 4.92	95.04 ± 5.58

**Table 4 foods-11-01843-t004:** Evaluation of matrix effect with comparison of calibration-curve slopes.

Analyte	Matrix	Slope	Slope Matrix/Solvent	%ME
ENR	Water	2040.14239	0.9826	98.3%
	Milk	2004.60237		
CIP	Water	1018.39131	0.9661	96.6%
	Milk	983.82133		
OFL	Water	1586.97198	0.9782	97.8%
	Milk	1552.37639		
PEF	Water	2363.71296	0.9502	95.0%
	Milk	2246.1182		
LOM	Water	1654.79423	0.9532	95.3%
	Milk	1577.41885		
NOR	Water	934.71011	0.9793	97.9%
	Milk	915.32412		
SAR	Water	1347.2961	0.9671	96.7%
	Milk	1303.02713		
DAN	Water	2089.56314	0.9528	95.3%
	Milk	1991.0296		
DIF	Water	1053.1916	0.9416	94.2%
	Milk	991.72708		

**Table 5 foods-11-01843-t005:** Comparison of the selected analytical parameters using the developed method and the reported analytical methods.

Extraction Method	Detection Method	Target	Recovery Rate (%)	LOD	LOQ	Linearity Range	References
HLB-SPE	LC-MS/MS	14	79.0~119.9	0.5~1.5 μg/kg	2.0~5.0 μg/kg	2.5~100.0 μg/L	[26]
IL-DLLME-MSPE	HPLC	3	81.2~109.0	1.5 μg/L	4.0~8.0 μg/L	4~1000 μg/L	[27]
96-well-based	IFA	1	90.0~100.0	4.0 μg/L	/	0.01~400 μg/L	[28]
MSPE	HPLC-DAD	7	78.9~119.0	0.010~0.046 μg/kg	/	0.05~200.0 μg/kg	[29]
MIP-SPE	HPLC	4	76.8~97.7	10.0~20.0 ng/mL	20.0~50.0 ng/mL	20~1000 ng/mL	[30]
MNP	LFIA	10	16.47~83.67	1.0~2.0 ng/mL	/	0.2~10 µg/m	[31]
IA-MEPS	HPLC	8	53.9~90.6	0.05~0.1 ng/g	0.15~0.3 ng/g	0.1~100.0 µg/mL	[32]
PFSPE	LC-MS/MS	9	80.64~95.26	0.31~0.91 ng/mL	1.03~3.03 ng/mL	1.0~100.0 ng/mL	This study

## Data Availability

Data is contained within the article.

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
