# Peer review of "Simultaneous Determination of Nine Quinolones in Pure Milk Using PFSPE-HPLC-MS/MS with PS-PAN Nanofibers as a Sorbent"

_foods, 2022, doi:10.3390/foods11131843_

Round 1

Reviewer 1 Report

The present manuscript entitled "Simultaneous Detection of Nine Quinolones in Pure Milk using PFSPE-HPLC-MS/MS with PS-PAN Nanofibers as Sorbent" by Lanlan Wei, Yanan Chen, Dongliang Shao, Jingjun Li (foods-1745604) describes the development and validation of a novel method for determination of nine quinolones in milk samples. The polyacrylonitrile-polystyrene (PAN-PS) composite nanofibers as selective sorbent were prepared and characterized. The packed-fiber solid-phase extraction (PFSPE) -based method with PAN-PS nanofibers and high-performance liquid chromatography-tandem mass spectrometry (HPLC-MS/MS) was used to detect and determine nine quinolones. The proposed method was characterized as sensitive and cost-effective.

The present article is written correctly and has a good structure; moreover, it has all the necessary parts. The article is interesting from an analytical; therefore, it should interest the reader. My current decision is a minor revision. More specific comments and observations are presented below.

1. Title. Only detection or determination too?

2. Page 1, line 10. “analyze” should be removed. “Analyze” is used for samples, but detection or determination for analytes (nine quinolones).

3. Sometimes, words are separated strangely with a hyphen, e.g., hu-mans (page 1, line 28). Please check it in the whole manuscript and improve it.

4. Introduction. Please provide more details about the methods used (page 1, lines 40-42) and extraction procedures (page 2, lines 46-52).

5. Section 2.1. Please indicate whether stock solutions were a mixture of nine quinolones or single?

6. Section 2.2. What was the injection volume? Has the temperature been stabilized?

7. Table 1. An explanation of "*" should be provided below the table.

8. Section 2.4. The parameters used for each measurement technique should be added.

9. Section 2.5. Is it possible to add more information about samples (number of samples, manufacturer)?

10. Figure 1. The description of the part with the syringe is very difficult to see. The numbers and the description under the syringe should be enlarged.

11. RSD expressed as a percentage is the coefficient of variation (CV).

12. Figure 2. Scale bars can be more visible. (a’) and (b’) should be enlarged. Now it's hard to read everything.

13. Page 6, line 174. You mentioned 1454, but figure 3 shows 1450.

14. The authors showed that recovery is characterised by very good values. I recommend two articles (Talanta 96 (2012) 39-43 and Talanta 117 (2013) 64-69) describing recovery in the context of the accuracy of analytical results. Values of recoveries in the commonly accepted range are not always consistent with accurate results. Additive effects cannot be detected with the use of the recovery test at all. What can be done in the event of strong interference effects? How would you deal with them? What types of interference effects could occur?

15. Page 11, line 293. There should be Figure 6.

16. Figure 6 is too small and completely invisible.

17. Does the developed method have limitations? It would be good to add a brief discussion of the limitations.

18. Conclusion. Please, emphasize clearly the advantages of the research carried out.

19 References. Please check with journal requirements. I have the impression that journal abbreviations should be improved in a few cases. I am not sure about using et al. also in a few cases.

I hope that the comments presented will help improve the article.

Reviewer 2 Report

The study aimed to develop a reliable method for detecting nine adulterants (quinolones) in milk using a new sorbent. It would be a remarkable finding and cost-efficient since simplifying the extraction to obtain the targeted analytes. However, several points need to be improved.

Keywords: 

- Please consider using another keyword different from the title to enhance the discoverability.

Introduction:

- Line 61: Please justify why the PAN and PS were chosen as the composite nanofiber for the SPE.

- Line 69: Please also mention how those parameters were evaluated.

Material and methods:

- Line 109: the stirring process was performed at which speed (rpm)?

Results and discussion:

- Line 179: it would be better if the IR spectra of PS are presented in this manuscript to prove this statement.

- Line 265: what ME stand for?

- Line 288-292: this procedure should be mentioned in the material and methods instead of this section.

- Line 294: what is the justification of this statement? since there were no references or evaluations of the standard method. Also please specify the standard method, numerous analytical methods are available for extraction and detection.

Overall:

- There are several words has not been written in a correct way (hu-man, be-cause, anti-biotics, indi-cate)

- Please name all the equation formulas along with the manuscript, for example, equation 1, equation 2, etc.
